# ROBUST GRAPH REPRESENTATION LEARNING VIA PREDICTIVE CODING

## ABSTRACT

Graph neural networks have recently shown outstanding results in diverse types of tasks in machine learning, providing interdisciplinary state-of-the-art performance on structured data. However, they have been proved to be vulnerable to imperceptible adversarial attacks and shown to be unfit for out-of-distribution generalisation. Here, we address this problem by introducing a novel message-passing scheme based on the theory of predictive coding, an energy-based alternative to back-propagation that has its roots in neuroscience. As both graph convolution and predictive coding can be seen as low-pass filtering mechanisms, we postulate that predictive coding adds a second efficient filter to the messaging passing process which enhances the robustness of the learned representation. Through an extensive set of experiments, we show that the proposed model attains comparable performance to its graph convolution network counterpart, delivering strictly better performance on inductive tasks. Most importantly, we show that the energy minimization enhances the robustness of the produced presentation and can be leveraged to further calibrate our models and provide representations that are more robust against advanced graph adversarial attacks.

## 1 INTRODUCTION

Extracting information from structured data has always been an active area of research in machine learning. This, mixed with the rise of deep neural networks as the main model of the field, has led to the development of *graph neural networks* (GNNs). These models have achieved results in diverse types of tasks in machine learning, providing interdisciplinary state-of-the-art performance in areas such as e-commerce and financial fraud detection (Zhang et al., 2022; Wang et al., 2019), drug and advanced material discovery (Bongini et al., 2021; Zhao et al., 2021; Xiong et al., 2019), recommender systems (Wu et al., 2021), and social networks (Liao et al., 2018). Their power lies in a message passing mechanism among vertices of a graph, performed iteratively at different levels of hierarchy of a deep network. Popular examples of these models are *graph convolutional networks* (GCNs) (Welling & Kipf, 2016), and *graph attention networks* (Veličković et al., 2017). Despite the aforementioned results and performance obtained in the last years, these models have been shown to lack robustness and to be vulnerable against carefully-crafted adversarial attacks (Zügner et al., 2018; Günnemann, 2022). They have in fact been proved to be vulnerable and susceptible to imperceptible adversarial attacks (Dai et al., 2018; Zügner & Günnemann, 2019; Günnemann, 2022) and unfit for out-of-distribution generalisation (Hu et al., 2020). This prevents GNNs from being used in critical tasks, where misleading predictions may lead to serious consequences, or maliciously manipulated signals may lead to the loss of a large amount of money.

More generally, robustness has always been a problem of deep learning models, highlighted by the famous example of a panda picture being classified as a gibbon with almost perfect confidence after the addition of a small amount of adversarial noise (Akhtar & Mian, 2018). To address this problem, an influential work has shown that it is possible to treat a classifier as an energy-based generative model, and train the joint distribution of a data point and its label to improve robustness and calibration Grathwohl et al. (2019). Justified by this result, this work studies the robustness of GNNs trained using an energy-based training algorithm called *predictive coding* (PC), originally developed to model information processing in hierarchical generative networks present in the neocortex (Rao & Ballard, 1999). Despite not being initially developed to perform machine learning tasks, recent works have been analyzing possible applications of PC in deep learning. This is motivated by inter-

esting properties of PC, as well as its surprising similarities with BP: when used to train classifiers, PC is able to approximate the weight update of BP on any neural network (Whittington & Bogacz, 2017; Millidge et al., 2021), and a variation of it is able to exactly replicate the weight update of BP (Song et al., 2020; Salvatori et al., 2022b). It has been shown that PC is able to train powerful image classifiers (He et al., 2016), is able to perform generation tasks (Ororbia & Kifer, 2022), continual learning (Ororbia et al., 2020), associative memories (Salvatori et al., 2021), reinforcement learning (Ororbia & Mali, 2022) and train neural networks with any structure (Salvatori et al., 2022a). In this work, we extend the study of PC to structure data, and show that PC is naturally able to train robust classifiers due to its energy-based formulation. To show that, we first show that PC is able to match the performance of BP on small and medium tasks, hence showing that the results on image classification (Whittington & Bogacz, 2017) extend to graph data, and then showing the improved calibration and robustness against adversarial attacks of models trained this way. Summarizing, our contributions are briefly as follows:

- We introduce and formalise a new class of message passing models, which we call *graph predictive coding networks* (GPCN). We show that these models achieve performance comparable with equivalent GCNs trained using BP in multiple tasks, and propose a general recipe to train any message-passing GNN with PC.

- We empirically show that GPCNs are less confident in their prediction, and hence produce models that are better calibrated than equivalent GCNs. Our results show large improvements in expected calibration error (ECE) and maximumum calibration error (MCE) on the Cora, Citesser, and Pubmed datasets. This proves the ability of GPCN to estimate the likelihood close to the true probability of a given data point and capacity to better capture uncertainty in its prediction.

- We further conduct an extensive robustness evaluation using advanced graph adversarial attacks on various dimensions: poisoning and evasion, global and targeted, direct and indirect. In these evaluations, GPCNs outperforms its GCNs counterpert on all kinds of evasion attacks and gains over $10\%$ improvement on poisoning attacks on the most corrupted graph data and obtain a better performance on various datasets than other complex methods that use attention mechanisms (Veličković et al., 2017), or tricks designed to make the model more robust (Zhu et al., 2019).

## 2 PRELIMINARIES

In this section, we review the general framework of message-passing neural networks (MPNNs) (Gilmer et al., 2017). Let us assume a graph $G = (V, E, X)$ with a set of nodes $V$, a set of edges $E$, and a set of attributes or properties of each node in the graph, described by a matrix $X \in \mathbb{R}^{|V| x d}$. The idea behind MPNNs is to begin with certain initial node characteristics and iteratively modify them over the course of $k$ iterations using information gained from neighbours of each node, hence message passing, according to a multilayer structure. Let the representation of a node $u \in V$ at layer $k$ be $\mathbf{h}_u^{(k)}$. This representation is then iteratively modified by as follows:

$$\mathbf{h}_u^{(k)} = \text{update}^{(k)}\left(\mathbf{h}_u^{(k-1)}, \text{ aggregate }^{(t)}\left(\left\{\mathbf{h}_v^{(k-1)} \mid v \in N(u)\right\}\right)\right) \tag{1}$$

where $N(u)$ is a set of neighbors of node $u$, and *update* and *aggregate* are differentiable functions (i.e., neural networks). The $aggregate$ function has to be a permutation invariant to maintain symmetries necessary when operating on graph data such as locality and invariance properties. In this work, we will mainly focus on graph convolutional networks (GCNs). Here, the aggregation function is a weighted combination of neighbour characteristics with predetermined fixed weights, and *aggregate* function is a linear transformation. We chose GCNs as they tend to be lightweight and scale conveniently for large graphs.

### 2.1 PREDICTIVE CODING NETWORKS

Predictive coding networks (PCNs) were first introduced for unsupervised feature learning (Rao & Ballard, 1999), and later extended to supervised learning (Whittington & Bogacz, 2017). Here, we

will describe a recent formulation, called PC graphs, that allows to use PC to train on graphs with any topology (Salvatori et al., 2022a). What results, is a message passing mechanism that is similar to that of GNNs, but with no multilayer structure. Let us consider a directed graph $G = (V, E)$, where $V$ is a set of $n$ vertices, and $E$ the set of directed edges. Every vertex $u$ is equipped with a value node $h_{u,t}$, that denotes the neural activity of the node $u$ at time $t$, and is a variable of the model, and every edge has a weight $w_{u,v}$. Every node has a *prediction* $\mu_{u,t}$, given by the incoming signals from other layers processed by an aggregation function (in practice, always the sum), and prediction error $\varepsilon_{u,t}$, given by the difference between the real value of a node and its prediction. More in detail, we have

$$\mu_{u,t} = \sum_{v \in p(u)} w_{v,u} f(h_{v,t}) \quad \text{and} \quad \varepsilon_{u,t} = h_{u,t} - \mu_{u,t}, \tag{2}$$

where $p(u)$ denotes the set parent nodes of $u$. As PC graphs are energy-based models, training happens through minimization of the global energy in each layer. This global energy $F_t$ is the sum of the prediction errors of the network:

$$F_t = \frac{1}{2} \sum_u (\varepsilon_{u,t})^2. \tag{3}$$

Learning happens in two phases, called inference and weight update. The inference phase is a message passing process, where the weights are fixed, and the the values are continuously updated according to neighbour information. Differently from GNNs, where the update rule is given, here it follows an objective, that is to minimize the energy of equation 3. This is done via gradient descent until convergence. The update rule is the following:

$$\Delta x_{u,t} \sim \partial \mathcal{E}_t / \partial x_{u,t} = -\varepsilon_{u,t} + f'(x_{u,t}) \sum_{v \in c(u)} \varepsilon_{v,t} w_{v,u}, \tag{4}$$

where $c(u)$ is the set of children vertices of $u$. To perform a weight update, we fix all the value nodes, and update the weights for one iteration by minimizing the same energy function via gradient descent as follows:

$$\Delta w_{i,j} \sim \partial \mathcal{E}_t / \partial \theta_{i,j} = \alpha \cdot \varepsilon_{i,t}^l f(x_{j,t}), \tag{5}$$

## 3 GRAPH PREDICTIVE CODING NETWORKS

We now propose *graph predictive coding networks* (GPCNs), obtained by using the multilayer structure of GNNs, and the learning mechanism of PC. To design a GPCNs, we introduce two different message passing mechanisms, and incorporate them inside the same training algorithm. Both rules are derived by minimizing the same energy function of equation 3 via gradient descent. First, we propose an inter-layer predictive coding layer that adds neural activities and prediction errors. Second, we introduce an intra-layer predictive coding layer, which adds PC mechanism within the aggregation mechanism with a goal of filtering the message being passed down from direct neighbors of a particular node.

**Inter-layer graph predictive coding module:** We follow the formulation of PG graphs that we described in section 2.1. For a node $\mathbf{u} \in V$ and a message passing layer $k$, we have neural activity state denoted as $\mathbf{h}_{u,t}^k$ and corresponding prediction-error state, $\varepsilon_{u,t}^k$. $t$ denotes the inference phase time step during energy minimization. The predicted representation, $\mu_{\mathbf{u,t}}^{\mathbf{k}}$, at layer $k$ is calculated as follows:

$$\mu_{\mathbf{i,t}}^{\mathbf{k}} = \text{update}^{(k)} \left( \mathbf{h}_u^{(k-1)}, \text{aggregate}^{(t)} \left( \left\{ \mathbf{h}_v^{(k-1)} \mid v \in N(u) \right\} \right) \right) \text{ and } \varepsilon_{u,t}^k = \mathbf{h}_{u,t}^k - \mu_{u,t}^k \tag{6}$$

The embedding of node $u$ are obtained through global energy minimization during inference stage $\mathbf{h}_{i,t}^k$ as we described in section 2.1.

**Intra-layer graph predictive coding module:** To devise an intra-layer GPCN, we similarly apply the predictive coding mechanism to neighbourhood aggregation stage, where the neural activity

Table 1: Test accuracy on transductive tasks.

| Method | Cora | CiterSeer | PubMed |
|---|---|---|---|
| GCN | $\mathbf{80.72 \pm 1.05}\%$ | $67.12 \pm 1.53\%$ | $\mathbf{77.1 \pm 1.45}\%$ |
| GPCN | $80.7 \pm 1.09\%$ | $\mathbf{67.26 \pm 1.28}\%$ | $76.2 \pm 2.44\%$ |

Table 2: F-1 score on inductive tasks.

| Method | Cora | CiterSeer | PubMed | PPI(Sup.) | PPI(Unsup.) |
|---|---|---|---|---|---|
| GCN | $\mathbf{80 \pm 0.41}\%$ | $67.64 \pm 1.14\%$ | $77.0 \pm 0.46\%$ | $76.45 \pm 0.39\%$ | $52.44 \pm 0.37\%$ |
| GPCN | $79.66 \pm 0.75\%$ | $\mathbf{69.68 \pm 0.37}\%$ | $\mathbf{77.12 \pm 0.47}\%$ | $\mathbf{78.31 \pm 0.47}\%$ | $\mathbf{54.41 \pm 0.31}\%$ |

state of neighboring nodes of node $u$ is denoted as $\mathbf{h}_{N(u)}^{(k)}$ and its corresponding prediction-error state and predicted state are denoted as $\varepsilon_{\mathbf{agg}_{u,t}^k}$ and $\mu_{\mathbf{agg}_{u,t}^k}$, respectively. The equations governing the dynamic of this model are the following:

$$h_{u,t}^k = \mathrm{update}^{(k)}\left(\mathbf{h}_u^{(k-1)}, \mathbf{h}_{N(u)}^{(k)}\right) \tag{7}$$

$$\mu_{\mathbf{agg}_{u,t}^k} = \mathrm{aggregate}^{\,(k)}\left(\left\{\mathbf{h}_v^{(k-1)} \mid v \in N(u)\right\}\right) \tag{8}$$

$$\varepsilon_{\mathbf{agg}_{u,t}^k} = \mathbf{h}_{N(u)}^{(k)} - \mu_{\mathbf{agg}_{u,t}^k}. \tag{9}$$

In the similar fashion, $\mathbf{h}_{N(u)}^{(k)}$ is not updated directly, rather, it is updated during inference stage. The illustration of this method on example graph is shown in the supplementary material. In what follows, we will test GPCNs on some standard benchmarks on both inductive and transductive tasks.

## 4 EXPERIMENTS

In this section, we perform extensive experiments to assess the performance of our proposed on common benchmark tasks, their calibration, and the ability of our energy-based models to counter graph adversarial attacks. More details on the experiment setup, a description to reproduce all the results presented in this section, and further results can be found in the supplementary material. We now provide some information about the models and datasets used, as well as a description of the baselines we will compare against. We refer to the supplementary material for any further detail needed to reproduce the experiments.

**Datasets.** Following many related works (Welling & Kipf, 2016; Veličković et al., 2017; Zügner et al., 2018; Zhu et al., 2019), we conduct experiments using the standard citation graph benchmark datasets: Cora, Citeseer and Pubmed. We also employ inherently inductive large-scale protein-to-protein interaction (PPI) (Hamilton et al., 2017; Veličković et al., 2017) dataset to validate the scalability of our model. Dataset statistics are summarised in Tab. 5 in the supplementary material.

**Baselines.** To evaluate the performance of our framework, we compare our method with other techniques. We largely compare against GCN (Welling & Kipf, 2016) as our proposed method is based on the same message-passing scheme (although it can be easily be extended to other message-passing scheme, as we hinted above). For fair comparison, GCN is trained in similar setting without the bag of tricks commonly used in deep learning such as dropout or batch-normalisation. However, we also refer to results from other papers for GCN and other methods that use attention mechanisms, namely Robust-GCN (RGCN) (Zhu et al., 2019) and Graph Attention Network (GAT) (Veličković et al., 2017). **RGCN** was developed to increase robustness of GCN by representing node embeddings as Gaussian distribution and employ distribution variance in attention aggregation mechanism to counter adversarial attacks by penalizing nodes with high variance. **GAT** uses attention layers to assign different importance to neighbouring nodes. In graph defense research, GAT is usually used

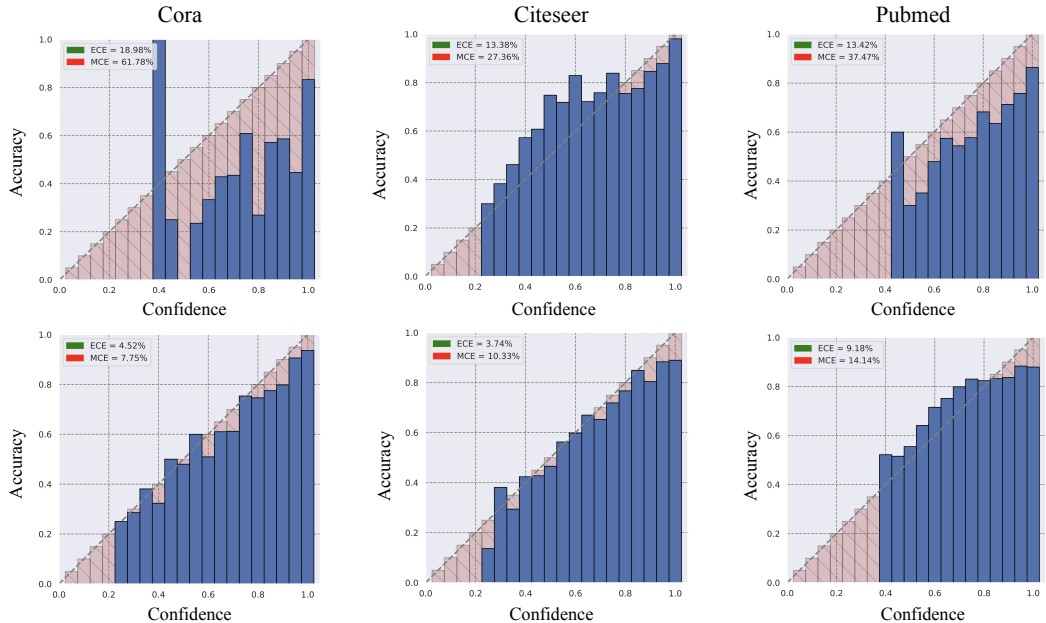

Figure 1: Reliability diagrams on multiple datasets. On top row, we have GCNs, on the bottom row, GPCNs. Inside every figure, we have the MCE and ECE of every model (the lower the better). Any deviation from the diagonal line indicates miscalibration. In all cases, PC outperforms standard GCNs trained with BP.

as a robustness baseline. Both GAT and RGCN are ideal to compare with our framework, since all rely on the GCN framework and are meant to mitigate drawbacks of GCN.

**Evaluation Metrics.** We are interested in learning calibrated models, that is the ability of a model to produce probability estimations that are accurate reflections of the correct likelihood an event. To do that, we employ scalar quantification metrics, such as expected calibration error (ECE) and maximum calibration error (MCE). The first, captures the notion of average miscalibration in a single number, and can be obtained by the expected difference between accuracy and confidence of the model. The second, quantifies worst-case expected miscalibration, and is hence crucial in safety and security-critical settings.

## 4.1 EXPERIMENTS ON GENERAL PERFORMANCE

To study where we stand in terms of performance against standard GNNs, we now test our newly proposed model against GCNs trained with BP on both transductive and inductive tasks. As shown in the extensive experiments reported in Tab. 1 and 2, this is indeed the case, especially for inductive tasks, where our method is almost always slightly better than BP. This is important, as it shows that PC has the potential to became an influential training method for GNNs. Furthermore, it is imperative that the performance of GPCNs be comparable to that of similar models trained with BP in order to make the improvements in robustness meaningful, that is the main goal of our work.

## 4.2 CALIBRATION ANALYSIS

Using the insights from the previous experiments and the fact that predictive coding framework can be seen as a generative-energy based model, we investigate calibration robustness our GPCN in comparison to GCN. To do that, we use reliability diagrams introduced in (Guo et al., 2017). Reliability diagrams are visual representation of model calibration that plots expected sample accuracy as a function of confidence. That is, if a model is perfectly calibrated, the diagrams would plot an identity function, any variation implies miscalibration. The reliability diagrams of both GCNs and GPCNs are in Fig. 1. GCNs are highly overconfident on most prediction confidence level on the Cora dataset, and highly under-confident on 0.4 prediction confidence. Conversely, on CiteSeer,

Table 3: Robustness against perturbations.

|  | GCN (Chen et al., 2021) | RGCN (Chen et al., 2021) | GCPN |
|---|---|---|---|
| Cora | $2.05 \pm 0.07$ | $2.79 \pm 0.10$ | $\mathbf{3.26 \pm 0.18}$ |
| CiterSeer | $1.98 \pm 0.12$ | $2.02 \pm 0.23$ | $\mathbf{2.73 \pm 0.08}$ |
| Pubmed | $1.14 \pm 0.02$ | $1.48 \pm 0.02$ | $\mathbf{4.21 \pm 0.32}$ |

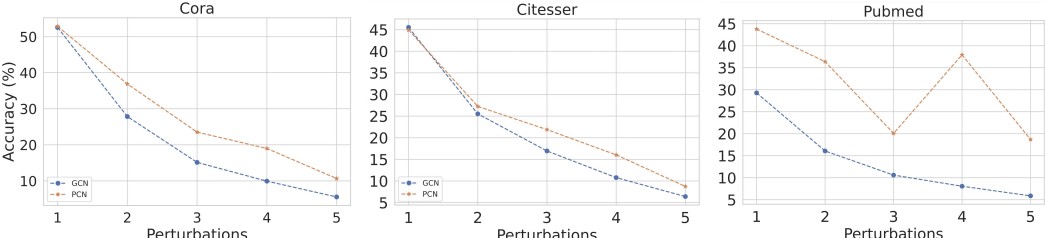

Figure 2: Robustness against direct attacks with respect to the number of perturbations on Cora (left), CiteSeer (centre), and Pubmed (right). In orange, GPCNs, in blue, GCNs.

GCNs tend to be under-confident on most predictions, which proves the miscalibration of GCN models. However, we see that our energy-minimised GPCN tends to approximate perfect calibration with very small variations on both Cora and Citeseer. The similar trend is seen on Pubmed and further evidence is supported by histograms of prediction confidence distributions provided in the supplementary material.

We have also quantified the calibration error, using ECE and MCE. The results can be found inside the plots in Fig. 1. In all cases, our model again outperforms GCNs. It is important to not only view this metrics in relative terms, but in terms of the fact that ECEs of GPCN model are closer to 0, which would be the perfect calibrated model. These results are interesting, as they show that our models can effectively quantify uncertainty, which is crucial in high critical settings. In the reported experiments, we have always used the same learning rate. However, it is known that different learning rates have different impact on calibration (Guo et al., 2017). To show that our results are robust under variations of the learning rate, we have provided plots of how ECE and MCE change over time with different learning rates in the supplementary material, as well as further details needed to reproduce the experiments.

### 4.3 EVASION ATTACKS WITH NETTACK

We now evaluate our model on one of the advanced graph adversarial evasion attacks, Nettack (Zügner & Günnemann, 2019). In targeted evasion attacks, the model parameters are kept fixed. Then, we employ Nettack, which uses a surrogate model trained on the same training set to attack selected victim nodes. Again, all experiment are repeated five times under different random seeds.

First, following the experimental setting of (Chen et al., 2021), we assess robustness against structural attacks. We randomly select 1000 victims nodes from both validation and test set. As in previous works (Zügner & Günnemann, 2019; Jin et al., 2020), the perturbations budget ranges from 1 to 5 and each victim nodes is attacked separately. We employ $\sum_{q=1}^{5} q \times p_q$ as holistic robustness metric where $q$ denotes the number of perturbations and $p_q$ is the classification accuracy corresponding to perturbation budget $q$ Chen et al. (2021). A larger value of this metric correspond to high robustness. The results are displayed in Table 3, where GPCN strictly performs better than GCN and R-GCN (note these results are based on BP model that even use drop-out and batch-normalisation), with the highest robustness found on Pubmed, being the largest dataset among the three. The result on each perturbation budget are also plotted in Fig. 2, and from the figure, we observe that GPCN outperforms GCN for all budgets, especially as for large number of perturbations.

Second, we perform a semi-qualitative analysis of robustness using classification margin and box-plots. Victim nodes are selected in similar fashion as in Nettack paper (Zügner & Günnemann,

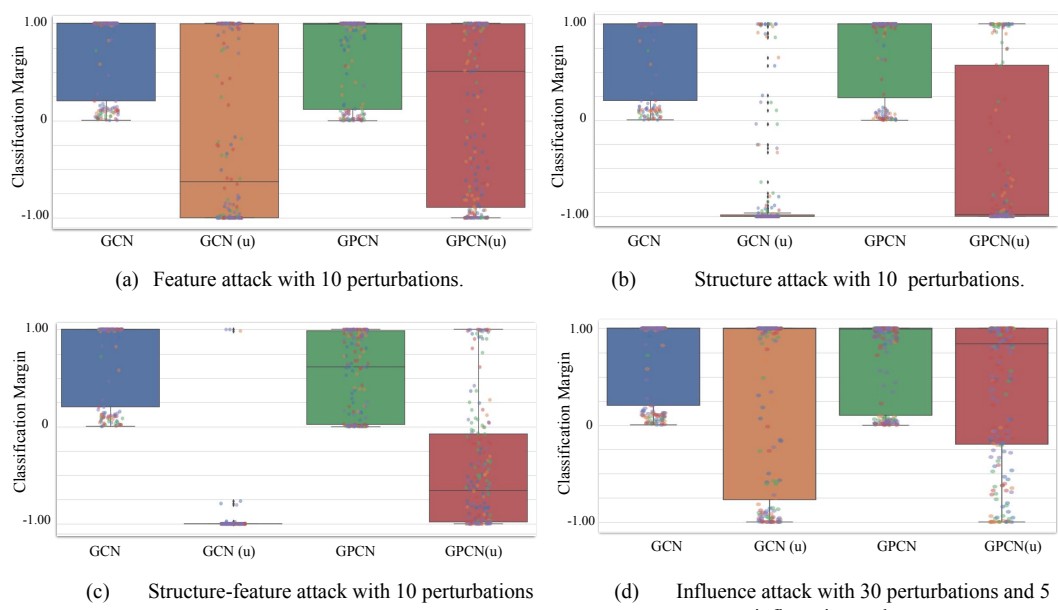

(a)  Feature attack with 10 perturbations.

(b)  Structure attack with 10 perturbations.

(c)  Structure-feature attack with 10 perturbations

(d)  Influence attack with 30 perturbations and 5 influencing nodes.

Figure 3: Classification margin diagrams on different types of attacks on Cora dataset. GCN(u) and GPCN(u) labels indicate the results of GCN and GPCN on clean/untempered graph data. Figure a) corresponds to adversarial attacks on node feature attacks; Figure b) corresponds to adversarial attacks on graph structures: edges; Figure c) corresponds to adversarial attacks on both structure and features; and Figure d) corresponds to indirect attack where adversarial attack targets neighboring nodes of a victim node

2019): we select 10 nodes with highest classification margins, 10 other nodes with lowest classification margins, and 20 nodes that are randomly selected from the test set. We then perform a range of attacks such as direct and indirect attacks and feature or/and structure attacks, and evaluate the robustness on varying perturbation budget. In the box-plots represented in Fig. 3, each point represents one victim node, and the color of point indicates the random seed on which an experiment is performed. The suffix '(u)' indicates the performance of a model on clean graphs. A more robust model is one that retain a higher classification margins after an attack.

1. **Structure and Feature attack:** Figure 3 (c) shows that with only 10 perturbation on neighbourhood structure and features of victim nodes, the classification margin of victim nodes collapses to $-1$ on the GCN model, while GPCN stays relatively robust with many victims nodes retaining positive classification margins, meaning that they were not adversarial affected by the attack. This trend is reflected for all numbers of perturbations $(2, 5, 10)$, and evidence is given in the supplementary material. In particular, when the number of perturbations is 2, the median of classification margin for GNN falls closer to $-1$, while GPCN protects around 50 percent of the victim nodes retaining positive classification margin after the attack.

2. **Feature attacks:** Since feature attacks do not affect GNNs as much as structure attacks, we use a high perturbation budget for features with perturbation numbers of 1, 5, 10, 30, 50 and 100. We observe a similar trends as the previous work where a small perturbation on features does not affect the model much. However, when perturbation rate becomes large, our GPCN displays unparalleled performance resisting the attacks. When perturbation rate is equal to 30 (see Figure 3 (a)), while GCN misclassifies around 70% of the victim nodes, GPCN is still able to classify more than 70% correctly after perturbation. The highly superior performance is observed when perturbation budget is increase to 100 (see the supplementary material), where GCN mislassifies all victims nodes, but GPCN is able to correctly classify most victims nodes in the upper quartile after the attack.

| Dataset | Ptb Rate (%) | GCN | GPCN-GCN | GAT (Jin et al., 2020) | RGCN (Jin et al., 2020) |
|---|---|---|---|---|---|
| | | | *Poisoning Attack with Mettack* | | |
| Cora | 0 | $82.87 \pm 0.75$ | $83.17 \pm 0.78$ | $\mathbf{83.97 \pm 0.65}$ | $83.09 \pm 0.44$ |
| | 5 | $76.4 \pm 0.88$ | $78.17 \pm 1.13$ | $\mathbf{80.44 \pm 0.74}$ | $77.42 \pm 0.39$ |
| | 10 | $67.98 \pm 0.99$ | $71.21 \pm 1.13$ | $\mathbf{75.61 \pm 0.59}$ | $72.22 \pm 0.38$ |
| | 15 | $60.3 \pm 1.73$ | $65.14 \pm 1.84$ | $\mathbf{69.78 \pm 1.28}$ | $66.82 \pm 0.39$ |
| | 20 | $50.31 \pm 1.69$ | $55.83 \pm 3.39$ | $\mathbf{59.94 \pm 0.92}$ | $\mathbf{59.27 \pm 0.37}$ |
| | 25 | $44.16 \pm 0.88$ | $54.27 \pm 7.25$ | $\mathbf{54.78 \pm 0.74}$ | $50.51 \pm 0.78$ |
| Citeseer | 0 | $72.43 \pm 0.05$ | $72.64 \pm 0.51$ | $\mathbf{73.26 \pm 0.83}$ | $71.20 \pm 0.83$ |
| | 5 | $71.53 \pm 0.43$ | $72.1 \pm 1.07$ | $\mathbf{72.89 \pm 0.83}$ | $70.50 \pm 0.43$ |
| | 10 | $68.59 \pm 0.65$ | $69.04 \pm 0.45$ | $\mathbf{70.63 \pm 0.48}$ | $67.71 \pm 0.30$ |
| | 15 | $65.02 \pm 1.16$ | $65.95 \pm 1.21$ | $\mathbf{69.02 \pm 1.09}$ | $65.69 \pm 0.37$ |
| | 20 | $53.77 \pm 0.73$ | $56.73 \pm 1.67$ | $61.04 \pm 1.52$ | $\mathbf{62.49 \pm 1.22}$ |
| | 25 | $57.49 \pm 2.13$ | $58.43 \pm 1.7$ | $\mathbf{61.85 \pm 1.12}$ | $55.35 \pm 0.66$ |
| PubMed | 0 | $85.37 \pm 0.06$ | $85.3 \pm 0.3$ | $83.73 \pm 0.40$ | $\mathbf{86.16 \pm 0.18}$ |
| | 5 | $81.4 \pm 0.12$ | $\mathbf{81.99 \pm 0.24}$ | $78.00 \pm 0.44$ | $81.08 \pm 0.20$ |
| | 10 | $79.73 \pm 0.27$ | $\mathbf{80.47 \pm 0.73}$ | $74.93 \pm 0.38$ | $77.51 \pm 0.27$ |
| | 15 | $77.03 \pm 0.11$ | $\mathbf{79.38 \pm 0.43}$ | $71.13 \pm 0.51$ | $73.91 \pm 0.25$ |
| | 20 | $75.59 \pm 0.26$ | $\mathbf{78.01 \pm 0.37}$ | $68.21 \pm 0.96$ | $71.18 \pm 0.31$ |
| | 25 | $73.34 \pm 0.19$ | $\mathbf{75.69 \pm 1.43}$ | $65.41 \pm 0.77$ | $67.95 \pm 0.15$ |

Table 4: Classification performance of the models under poisoning global attack with Metattack.

3. **Structure attacks**: GPCN also consistently outperforms GCN under structure-only attacks as it can be observed in the Figure 3 (b) under 10 perturbations. More figures under different number of perturbations $(1, 2, 3, 5, 10)$ are provided in supplementary material.

4. **Indirect Attacks:** For indirect attacks, we choose 5 influencing/neighboring nodes to attack for each victims node. We observe a similar trend that was found in the original Nettack paper (Zügner et al., 2018). We found that the influence or indirect attacks do not affect GNNs as much as other types of attack as it can be witnessed from the box plot in Figure 3 (b). However, we found that GPCN consistently outperforms GCN model. Most importantly, we found that GPCN is hardly affected on all perturbation budget as the lower quartiles of box all box plots stay in positive half of classification margin for all attacks (see the supplementary material).

## 4.4 GLOBAL POISONING ATTACKS USING META LEARNING (METTACK)

Finally, we perform global poisoning attacks using Mettack technique (Zügner & Günnemann, 2019). In poisoning attacks, only the training data are corrupted and are tempered with in a manner that renders the target model fail to learn. This is the most common type of graph attacks in the real world, as malicious individual can change the training data, but do not have access to the parameters of the model (Jin et al., 2021). As Mettack has several variants, we use the same setting as in the Pro-GNN paper (Jin et al., 2020) and employ the most destructive variant known as Meta-Self on Cora and CiteSeer, and apply A-Meta-Self (approximate faster version of Meta-Self) on Pubmed due to computation limitations. The perturbation rate is varied from $0\%$ to $25\%$ with a step of 5 and the results are reported in Table 4, where we also compare with the results obtained in the Pro-GNN work. Note that the reason the accuracy for 0 perturbation is different from one we reported earlier, here we use only the largest component of a graph instead of using all nodes.

As it can be seen from Table 4, GPCN consistently performs better than GCN on all datasets with more than $10\%$ increase in robustness on Cora dataset when perturbation rate is equal to $25\%$. GPCN also outperforms other methods under various perturbation rates, particularly outperforming all methods on all perturbation rate on PubMed with more than $7\%$ improvement over both GAT and RGCN when perturbation rate is $25\%$. This is interesting because both GAT and RGCN were trained using dropout and batch-normalisation and both use attention mechanism in their aggregation process. In particular, RGCN was specially developed to improve robustness of GCN.

## 5 RELATED WORK

**Adversarial attacks on graphs.** The recent revelation of lacks of robustness of the current graph learning methods inspired a body of work that attempts to enhance the robustness of graph machine learning. Those techniques can generally be classified into three categories: robust representation, robust detection, robust optimisation (Ma et al., 2021). Robust representation entails techniques that seek to map graph representation into a resilient embedding space by minimising the loss objective function of anticipated worst-case perturbation approximations such as robustness certificates (Bojchevski & Günnemann, 2019) and known adversarial samples (Xu et al., 2019). Robust detection techniques, on the other hand, recognise that the dearth of robustness of GNNs stems from local message-passing aggregation phase; thus, they selectively choose which neighbourhood nodes to include in aggregation based on some properties. Popular techniques in this category include Jaccard method (Wu et al., 2019), which removes edges of some nodes whose Jaccard similarity below a certain threshold, and singular value decomposition method (Entezari et al., 2020), which preprocesses a graph by generating a low-rank approximation of it. Finally, robust optimisation is concerned with regularisation techniques that avoid extreme embeddings. GCN-LFR (Low-Frequency based Regularisation) (Chang et al., 2021) adopts a robust co-training paradigm that derive the robustness from the eligible low-frequency components, while MedianGCN (Chen et al., 2021) leverages robust aggregation functions (i.e., the median and trimmed mean) that ignore outliers based on a breakdown point characterisation. MedianGCN is closely similar SMGCN (Geisler et al., 2020) that introduces Soft Medoid function as a message-aggregation method to produce robust representation. Robust-GCN(RGCN) (Zhu et al., 2019) embeds node representation as a Gaussian distribution and utilises variance-based attention mechanism during neighbourhood message aggregation phase.

**Energy-based Models (EBMs).** Although there is a considerable interest in integrating energy-based view into deep learning (Xie et al., 2016; Nijkamp et al., 2019; Grathwohl et al., 2019; Song & Kingma, 2021), only a handful of work has transferred it to graph machine learning (Di Giovanni et al., 2022). Here, most lines of research have largely concentrated on graph generation tasks with models such as GNN-EBMS (Liu et al., 2020), and GraphEBM (Liu et al., 2021). Only one nascent work has recently attempted to expand the GCN classifier to an energy-based model named GCN-JEMO (Shin & Dharangutte, 2021). GCN-JEMO derives its energy from graph properties and was demonstrated to achieve comparable discriminative performance to classic GCN but with increased robustness. In contrast to our work, GCN-JEMO relies on non-standard training method and is not tested on adversarial attacks.

**Predictive Coding.** Recently, many works have been developed, that use PC to address machine learning problems. A first example is computer vision, where recent works have performed image classification with simple experiments on MNIST (Whittington & Bogacz, 2017), or more complex ones on Imagenet (He et al., 2016). Other examples are image generation (**?**), associative memories (Salvatori et al., 2021), continual learning (Ororbia et al., 2020), reinforcement learning (Ororbia & Mali, 2022), NLP (Pinchetti et al., 2022). We conclude by referring to a more theoretical direction, that is Friston's free energy principle and active inference (Friston, 2010; Friston et al., 2006; 2016).

## 6 DISCUSSION

In this work, we have explored a new framework to perform machine learning on structured data, inspired from the neuroscience theory of PC. First, we have defined the model, and then we have shown that it is able to reach competitive performance in both inductive and transductive tasks, with respect to similar models trained with BP. We have then tested this framework on robustness tasks, with extensive results showing that simply training GNNs using PC instead of BP, results in models that are better calibrated, and more robust against adversarial attacks.

As we have used the original formulation adapted to GNNs, with no further effort put in increasing the robustness of the trained models, future work should focus on scaling up the results of this paper to large-scale models, and research on variations of the proposed framework that make these models even more robust. Furthermore, while here we have only considered GCNs, it would be interesting to study different models, such as graph attention networks, trained using PC. More generally, this work shrinks the gap between computational neuroscience and machine learning, by showing that biologically plausible methods are able to reach competitive performance on complex tasks.

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

## A    DETAILS ON EVALUATION METRICS

In this section, we explain the metrics used in the main body of this work in more detail. We employ scalar quantification metrics, such as expected calibration error (ECE) and maximum calibration error, together with visual tools, such as reliability diagrams and confidence distribution histograms (Guo et al., 2017) and classification margin diagrams (Dai et al., 2018) to evaluate model calibration.

The term 'confidence calibration' is used to describe the ability of a model to produce probability estimations that are accurate reflections of the correct likelihood an event, which is imperative especially in real-world applications. Considering a $K$-class classification task, let $X \in \mathcal{X}$ and $Y \in \mathcal{Y} = \{1, \ldots, K\}$ be input and true ground truth label random variables, respectively. Let $\hat{Y}$ denote a class prediction and $\hat{P}$ be its associated confidence, i.e. probability of correctness. We would like the confidence estimate $\hat{P}$ to be calibrated, which intuitively means that $\hat{P}$ represents a true probability. The perfect calibration can be described as follows:

$$\mathbb{P}(\hat{Y} = Y \mid \hat{P} = p) = p, \quad \forall p \in [0, 1] \tag{10}$$

**ECE** captures the notion of average miscalibration in a single number, and can be obtained by the expected difference between accuracy and confidence of the model: $\mathbb{E}_{\hat{P}}[|\mathbb{P}(\hat{Y} = Y \mid \hat{P} = p) - p|]$

**MCE** is much crucial in safety-, security-critical settings, as it quantifies worst-case expected miscalibration: $\max_{p \in [0,1]} |\mathbb{P}(\hat{Y} = Y \mid \hat{P} = p) - p|$.

**Reliability diagrams** are visual representation tools for model calibration as equation 10 is intractable because $\hat{P}$ is a continuous random variable. They characterise the average accuracy level inside a points that feels into a given confidence level bin (see  for more details on formulation of the the above metrics).

**Classification margin** is simply the difference between the model output probability of the ground truth class and the model probability of the predicted most-likely class, i.e.,the probability of the best second class. Thus, this metric is between $-1$ and $1$, where values close to $-1$ indicate that a model is overconfident in wrong predictions and values closer to $1$ indicate that the model is confidence in its correct prediction. In our reported class margin diagrams, we average these values over many samples and repeated trials with different random seeds to draw box-plot diagrams of our results.

## B    REPRODUCIBILITY

We use standard splits on all datasets. We report the average results of 5 runs on different seeds; the hyper-parameters are selected using the validation set. Following the results from original papers on baseline models, we evaluate our model on 2 GNN layers. The models are trained for 300 epochs on citation graphs and 50 epochs on PPI. We also use Adam optimiser. The reported results on calibration analysis were performed on initial learning rate of 0.001 for both GCN and GPCN as it provided best performances for both models. For adversarial attacks, we set initial learning rate to 0.01 and epochs to 200 to compare our results to other works. For general performance experiment, we use grid search for hyper-parameters as described in Table 6. Inductive tasks were trained using GCN version of GraphSage (Hamilton et al., 2017) with neighborhood sample size of 25 and 10 for first and second GNN layer, respectively (see Hamilton et al. (2017) for more details).

Our experiments are performed using Pytorch Geometric library Fey & Lenssen (2019). In order to do an extensive experiment, we build on GraphGym You et al. (2020), a research platform for designing and evaluating GNNs, and we seamlessly integrate it with predictive coding learning algorithm. In addition, we employ another Pytorch library for adversarial attacks and defenses known as Deeprobust Li et al. (2020) for various type of adversarial attacks we perform on graphs.

## B.1 DATASETS

Table 5: An overview of the data sets used in our experiments

|  | Cora | Citeseer | Pubmed | PPI | Reddit |
|---|---|---|---|---|---|
| Type | Citation | Citation | Citation | Protein interaction | Communities |
| #Nodes | 2708 | 3327 | 19717 (1 graph) | 56944 (24 graphs) | 232965 (1 graph) |
| #Edges | 5429 | 4732 | 44338 | 818716 | 114615892 |
| #Features/Nodes | 1433 | 3703 | 500 | 50 | 602 |
| #Classes | 7 | 6 | 3 | 121(multilabel) | 41 |
| #Training Nodes | 140 | 120 | 60 | 44906(20 graphs) | 153431 |
| #Validation Nodes | 500 | 500 | 500 | 6513 (2 graphs) | 23831 |
| #Testing Nodes | 1000 | 1000 | 1000 | 5524 (2 graphs) | 55703 |

| Parameter Type | Grid |
|---|---|
| values nodes update rate | $0.05, 0.1, 0.5, 1.0$ |
| Weight update learning rate | $1e-2, 1e-3, 1e-4, 1e-5$ |
| Number of GNN Layer | $2, 4$ |
| Inference steps,T, | 12, 32, 50, 100 |
| PC synaptic weight update rate | at the end of ,T, inference steps, and at every inference step |
| aggregation functions | sum, add, max |
| Graphsage sampling | 10, 25 for first and second GNN layer respectively |

Table 6: Hyper-Parameter Search

## C  CALIBRATION ANALYSIS: CONFIDENCE IN PREDICTION

As deep networks tend to be overconfident even if they are wrong, we compared the confidence distribution of GPCNs with GCNs in Figure 4, 5, and 6, where confidence is the maximum of the softmax of the model output. The result in this section correspond to the reported results in the body of the paper on calibration analysis in Section 4.2. Both model are run for 300 epochs using the same parameters (i.e., learning rate on weights equal to 0.001) and we select the best model based on validation set for GCN. For GPCN, we select the best model based on best accuracy on validation set as well as lowest energy on training set as the energy minimization can be interpreted as likeli-hood maximisation. We consider energy while selecting the best model for evaluation because we discovered a high correlation between energy and robustness as we will demonstrate in the following section (see Figure 7). Interestingly, the results on prediction distribution on provide another dimension to communicate the same results we witness in Section 4.2 using reliability diagrams. We see that on the prediction distribution on Cora dataset in Figure 6, GPCNs are relatively less confident in their predictions, while GCN model is overly confident. Figure 5 on Citeseer dataset similarly here shows that most prediction confidence of GCN model are less than 0.5, showing that GCN models are overly under-confident as we saw in the body of the paper. GPCNs models on the other hand provide a well behaved prediction confidence distribution on Citeseer. demonstrate that either GCN either is under-confident despite high performance accuracy or over-confident in the manner that is disproportional to its performance accuracy.

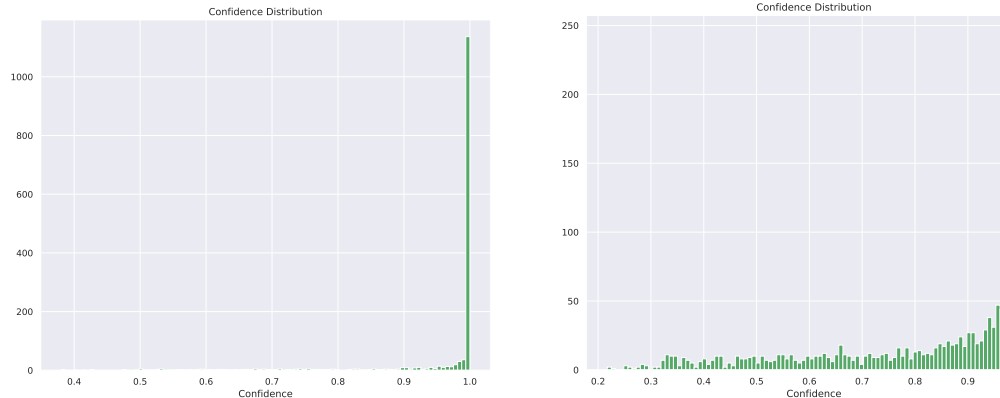

Figure 4: Histogram of prediction confidence distribution on Cora dataset. The x-axis indicates the confidence of the model on the samples, and y-axis is the count on a normal scale of data points that fall into a given confidence bin. Left: GCN. Right: GPCN.

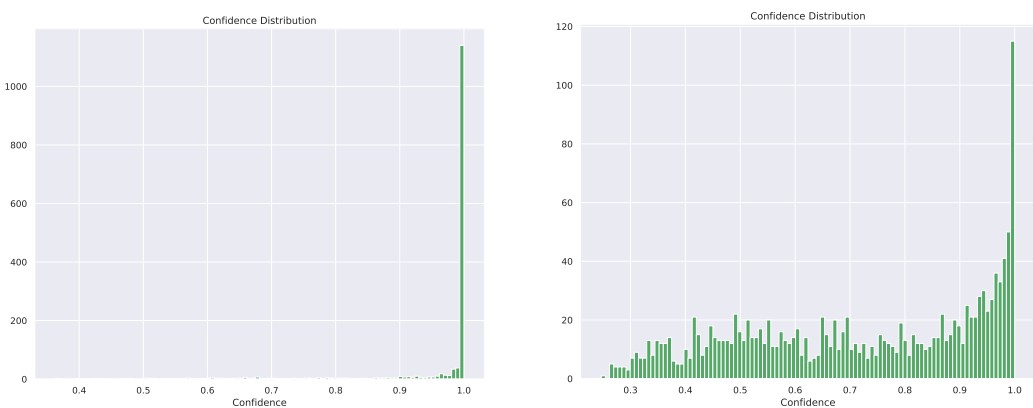

Figure 5: Histogram of prediction confidence distribution on Citeseer dataset. The x-axis indicates the confidence of the model on the samples, and y-axis is the count on a normal scale of data points that fall into a given confidence bin. Left diagram is for GCN while right diagram is ouput of GPCN.

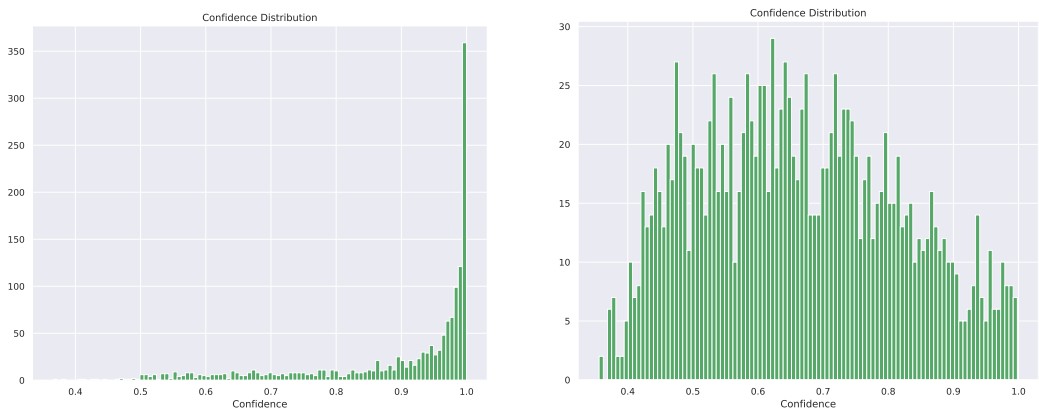

Figure 6: Histogram of prediction confidence distribution on PubMed dataset. The x-axis indicates the confidence of the model on the samples, and y-axis is the count on a normal scale of data points that fall into a given confidence bin. Left: GCN. Right: GPCN.

### C.1 Calibration Strengthening through Energy Minimization

We observed a high positive correlation between calibration and training energy level, thus, we further investigated the role of energy to robustness and calibration of our learned representation. Unlike the standard predictive coding network, we observed, GPCN requires several inference steps to reach lowest training energy possible, i.e., this can seen as reaching local optimal of likelihood maximisation function. More importantly, we also observed that the lower the energy the better the calibration is, i.e., the better the model can estimate uncertainty in its prediction. Figure 7 on Cora dataset and Figure 8 on CiteSeer dataset showcase this correlation. The plots on the left show that inference steps correlate to the energy level, i.e., the longer the inference is, the more likely that the model converge to a lower energy. The middle diagram and right diagrams, similarly, present the correlation between the inference steps and ECE and MCE, respectively, which from left diagrams, implies the correlation of energy level and ECE and MCE. We see that the lower the energy the better the calibration performance reached.

As it has been shown that calibration can be affected by learning rates Guo et al. (2017), we track the ECE and MCE throughout training, and we see that as the lower the learning rate, the better and more stable calibration GPCN is able to attain based on ECE and MCE metrics (see Figure 9 and 10).

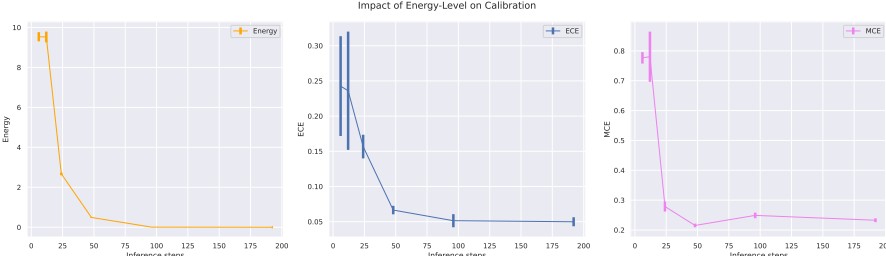

Figure 7: Evaluation of the impact of energy on calibration performance on Cora dataset (learning rate = 0.001). Left: Demonstrates that increasing inference steps leads to lower energy. Middle: Shows that the lower energy level determines expected error(ECE). Right: Also showcase, the correlation between energy and maximum calibration error(MCE)

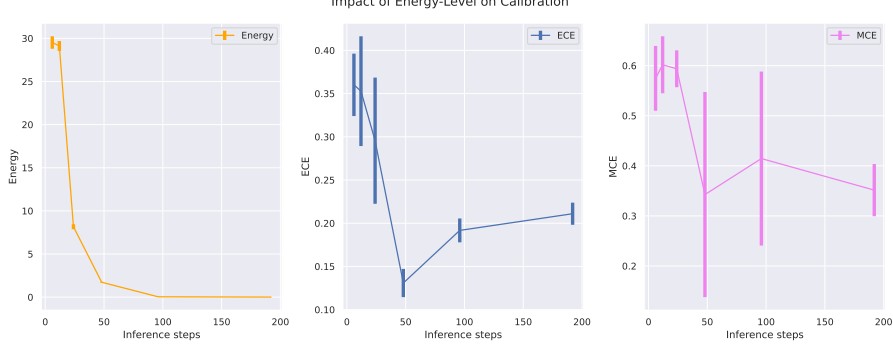

Figure 8: Evaluation of the impact of energy on calibration performance on CiteSeer dataset. Left: Demonstrates that increasing inference steps leads to lower energy. Middle: Shows that the lower energy level determines expected error(ECE). Right: Also showcase, the correlation between energy and maximum calibration error(MCE)

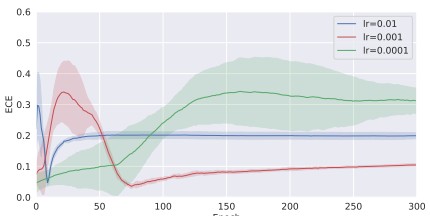 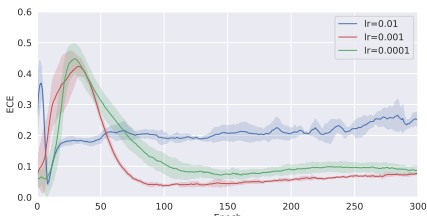

Figure 9: Evolution of expected calibration error (ECE) on test set during training on various learning rates (lr). Left: GCN. Right: GPCN.

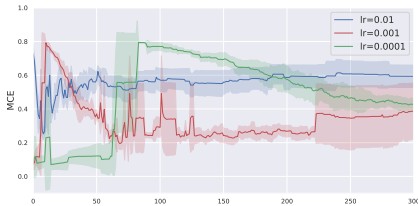 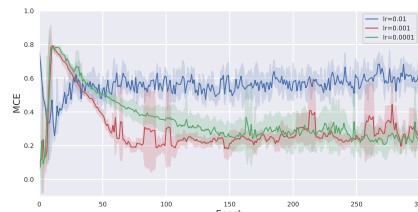

Figure 10: Evolution of maximum calibration error (MCE) on test set during training on various learning rates (lr). Left: GCN. Right: GPCN.

## D  GPCN ARCHITECTURE

Using a simple graph in Figure 11, here, we demonstrate the idea of graph predictive coding as opposed to standard graph neural networks. In typical graph convolution network (GCN) Welling & Kipf (2016) models, the node representation is obtained by the recursive aggregation of representations of its neighbours, after which a learned linear transformation and non-linear activation function are applied. After the $k_t h$ round of aggregation (with $k$ denoting the number of GCN layer), the representation of a node reflects the underlying structure of its nearest neighbours within $k$ hops. Note that the GCN is one of the simplest GNN model, as the update function equates to only neighbourhood aggregation, that is why in Figure 12(left) we only depict the aggregate function as it captures the update function altogether.

Our GPCN model (see Figure 12(right)) differs from the standard GNN in three aspects. First, node representation are not a mere result of neighborhood aggregation. Rather, each node has unique a neural state that is updated through energy-minimization using the theory of predictive coding described in the main body of this work. Specifically, each neighborhood aggregation at each hop, k,, pass through a predictive coding module that predicts the the incoming aggregated neighborhood representation. Second, GPCN has a different concept of what neighborhood messages are (see Figure **??**). Rather than transmitting raw messages, it instead, forward residual error of the difference between the predicted representation and the aggregation which reduce the dynamic range of the message being transmitted hence acting as low pass filter. Lastly, unlike the standard GNN that are trained using BP, where the update of weights corresponding to a given neighborhood are dependent, which create a large computation graphs, GPCN learning rules are local and the model weight are updated through energy minimization as we described in the methodology section.

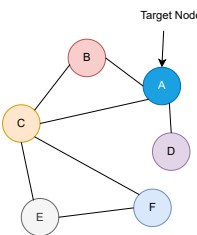

Figure 11: Example of graph that we use to illustrate the architecture.

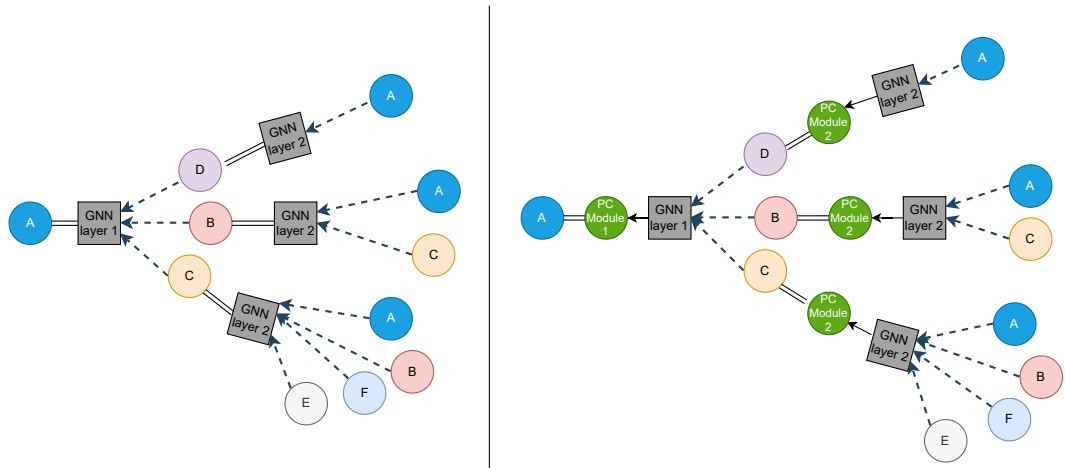

Figure 12: Distinction of Messaging propagation between standard GNNs (Left) and inter-layer GPCN (Right).

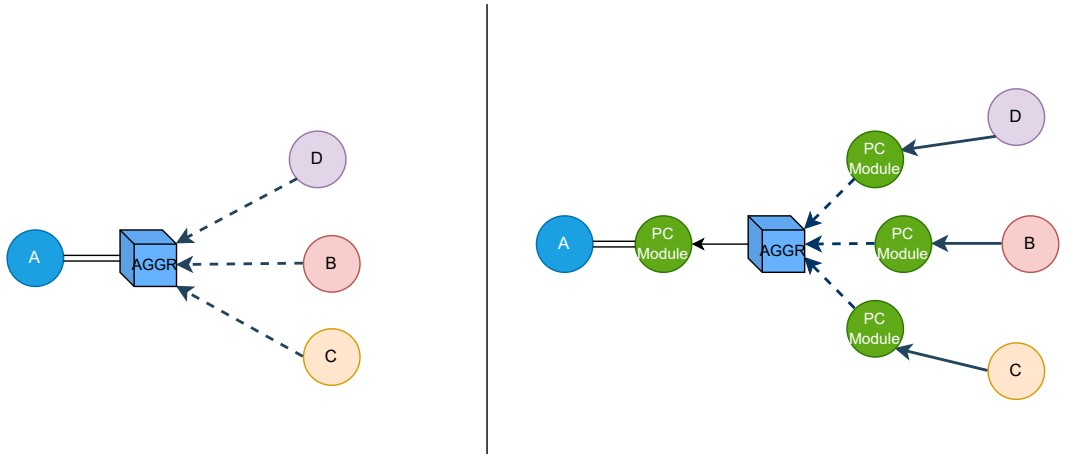

Figure 13: Distinction of the difference between message aggregation between standard GNNs and GPCN. Left: standard aggregation method, Right: intra-layer GPCN.

# E  ADDITIONAL RESULTS ON EVASION ATTACKS WITH NETTACK

Due to the page constraint, here we provide additional results we could not include in the main body of this paper. In particular, the folowing plots demonstrate how both GCN and GPCN model

perform under various perturbation budgets on the four types of attacks namely feature, structure, feature-structure, and indirect attacks.

**1) Structure and Feature attack:**

Figure 14 show that with only 2 perturbation on neighbourhood structure and features of victim nodes, the median classification margin approaches -1 on GCN model, while GPCN stays relatively robust and with more robustness on lower energy model(PCx3) where most of the victims nodes have positive classification margins, or in other worlds, they are not adversarially affect by the attack. This trend is even pronounced when perturbation rate is increase to five(Figure 15) and ten (Figure 16) where, except for outliers, margin of classification of all victims nodes falls to $-1$ for both GCN and PC model PCx3 stay lately more robust.

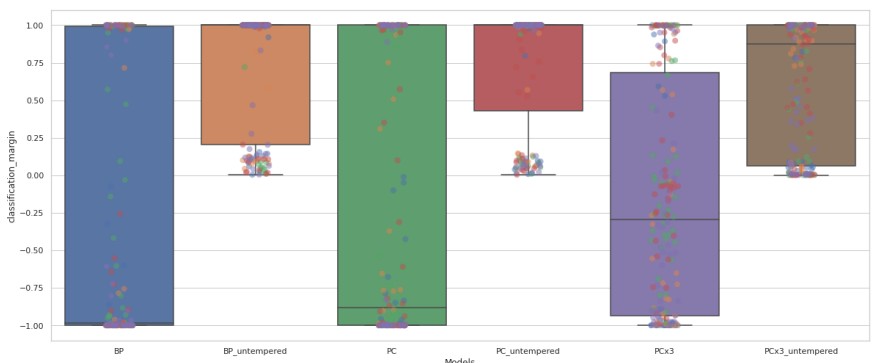

Figure 14: Classification margin diagram of targeted attack on both features and structure when perturbation rate equal to 2. On x-axis BP indicates GCN model, PC denotes GPCN model trained of 12 inference steps, and PCx3 indicate GPCN trained using 36 inference steps, hence achieves lower training energy.

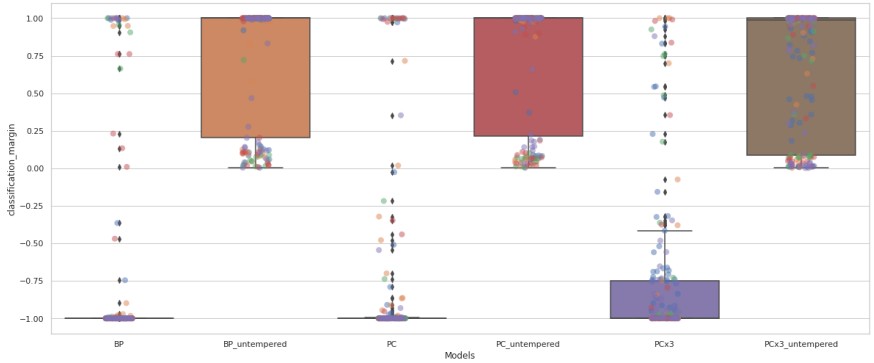

Figure 15: Classification margin diagram of targeted attack on both features and structure when perturbation rate equal to 5

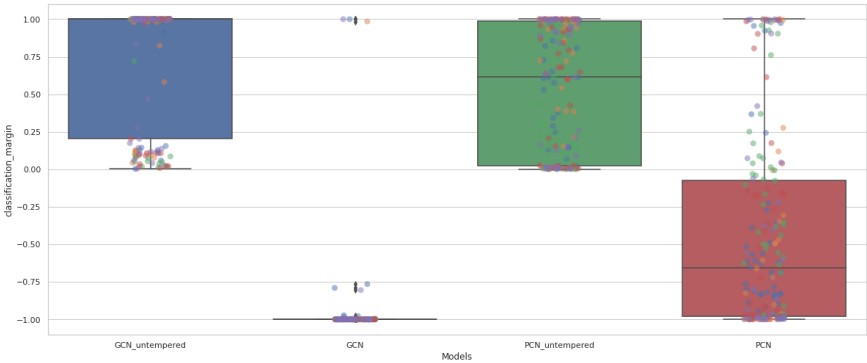

Figure 16: Classification margin diagram of targeted attack on both features and structure when perturbation rate equal to 10

**2) Feature attacks**. Since feature attacks do not high affect GNNs as much as structure attacks, we peform, large corruption of features with perturbation rate of 1, 5, 10, 30, 50 and 100. We observe a similar trends where with a small perturbation on features does not affect the model, however, when perturbation rate become large our GPCN display unparalleled performance resisting the attacks. When perturbation rate is equal to 30 (see Figure 20, while GCN misclassifies around 70% of the victim nodes, GPCN is still able to classify more than 70% correctly after perturbation. The highly superior performance is observed when perturbation rate is increase to 100 in Figure 22, GCN mislassifies all victims nodes, while GPCN still classify correctly those nodes with most victims nodes in the upper quartile having positive classification margins.

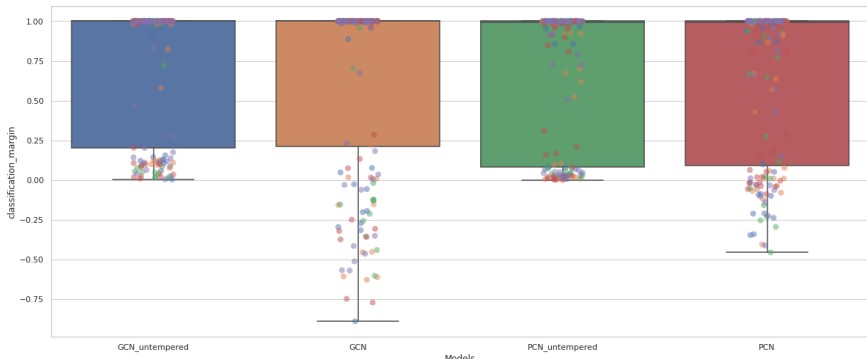

Figure 17: Classification margin diagram of targeted attack on features when perturbation rate equal to 1

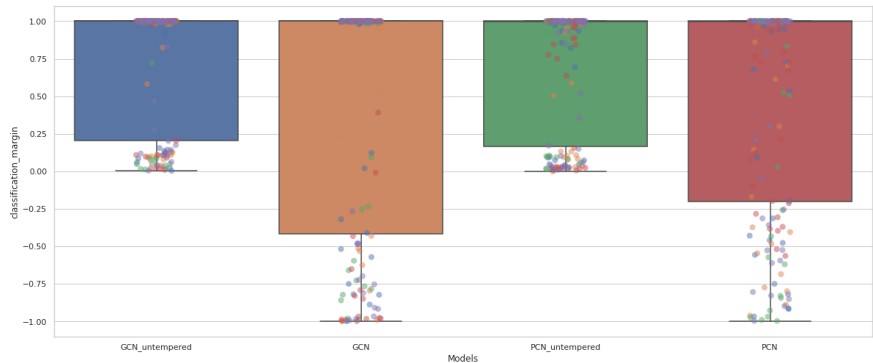

Figure 18: Classification margin diagram of targeted attack on features when perturbation rate equal to 5

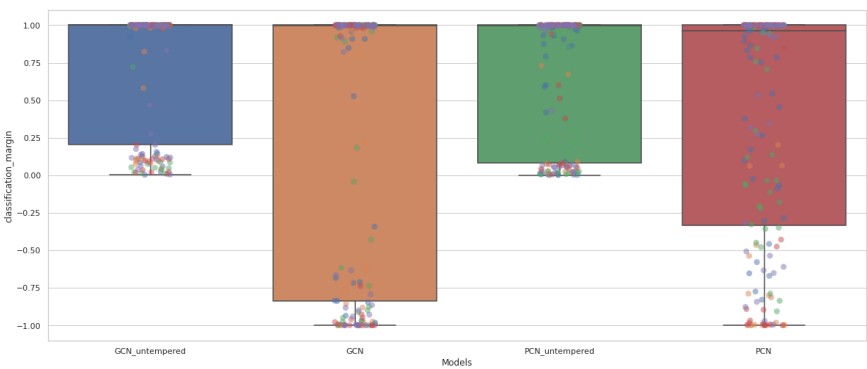

Figure 19: Classification margin diagram of targeted attack on features when perturbation rate equal to 10

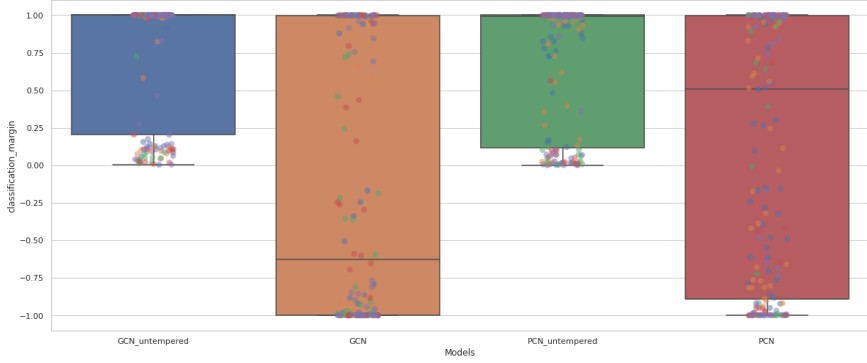

Figure 20: Classification margin diagram of targeted attack on features when perturbation rate equal to 30

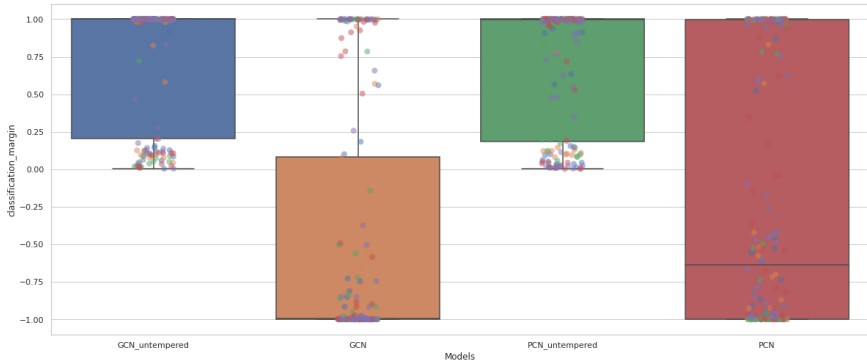

Figure 21: Classification margin diagram of targeted attack on features when perturbation rate equal to 50

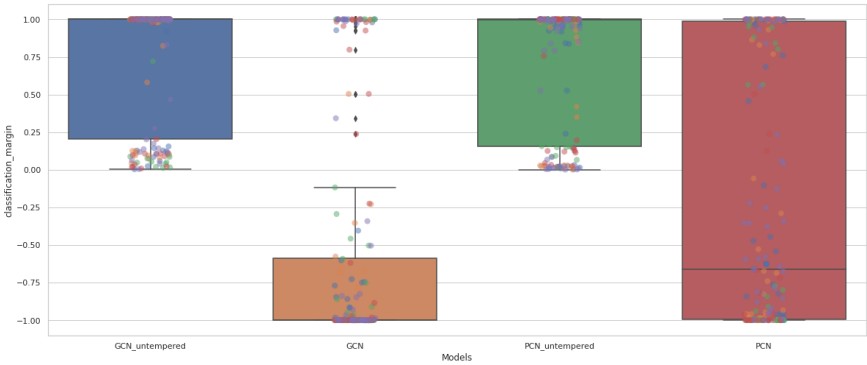

Figure 22: Classification margin diagram of targeted attack on features when perturbation rate equal to 100

**3) Structure attacks**: GPCN also consistently outperforms GCN under structure-only attacks on number perturbation equal to 1, 2, 5 and 10 as it can be observed in Figure 23,24,25,26, and 27.

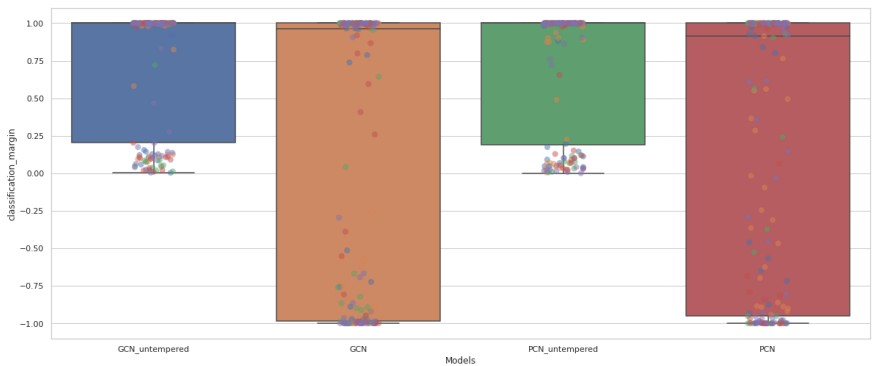

Figure 23: Classification margin diagram of targeted attack on structure when perturbation rate equal to 1

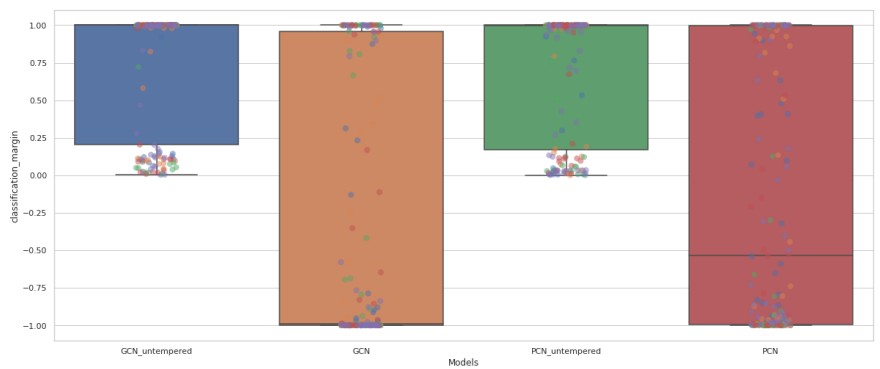

Figure 24: Classification margin diagram of targeted attack on structure when perturbation rate equal to 2

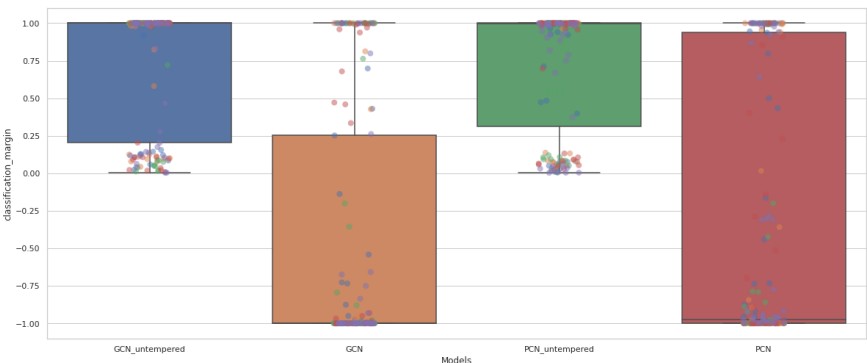

Figure 25: Classification margin diagram of targeted attack on structure when perturbation rate equal to 3

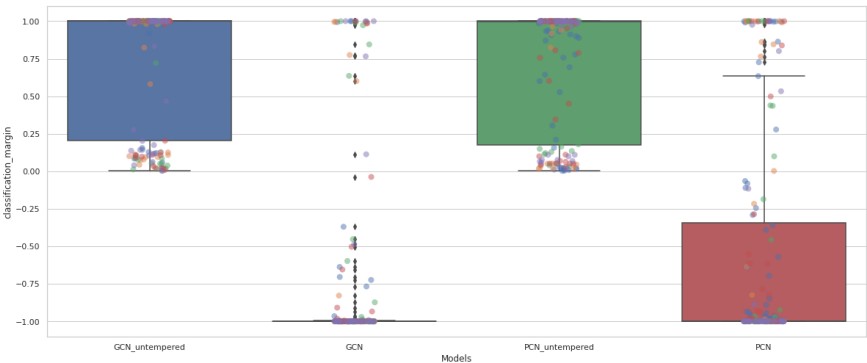

Figure 26: Classification margin diagram of targeted attack on structure when perturbation rate equal to 5

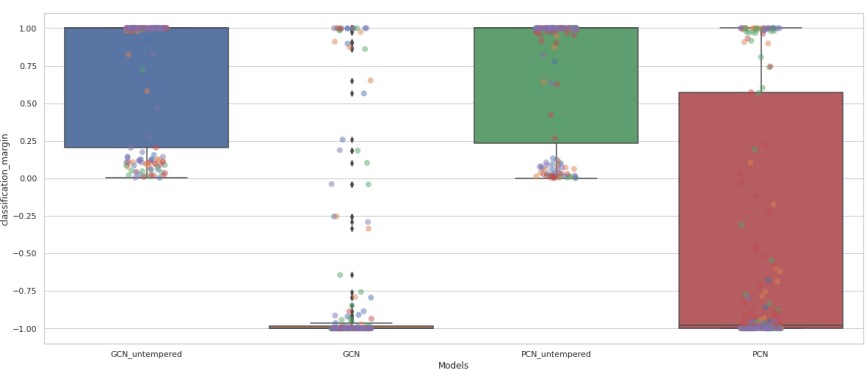

Figure 27: Classification margin diagram of targeted attack on structure when perturbation rate equal to 10

**4) Indirect Attacks**:

For indirect attacks, we choose 5 influencing/neighboring nodes to attack for each victims node. We observe a similar trend that was found in the Nettack paperZügner et al. (2018). We found that the influence or indirect attacks do not affects GNNs as much as other attacks as it can be witnessed from the box plots below. However, we also found that GPCN, especially one with smaller inference steps consistently outperforms all models, but all GPCN models are strictly better than GCN under all perturbations.

Note for Figure 28, 29, and 30, on x-axis, BP indicates GCN model, PC denotes GPCN model trained of 12 inference steps, and PCx3 indicate GPCN trained using 36 inference steps, hence achieves lower training energy. The suffix 'untempered' indicates the performance of model on clean graph. To interpret the plots, a more robust model is one that retain a higher classification margins after the attacks.

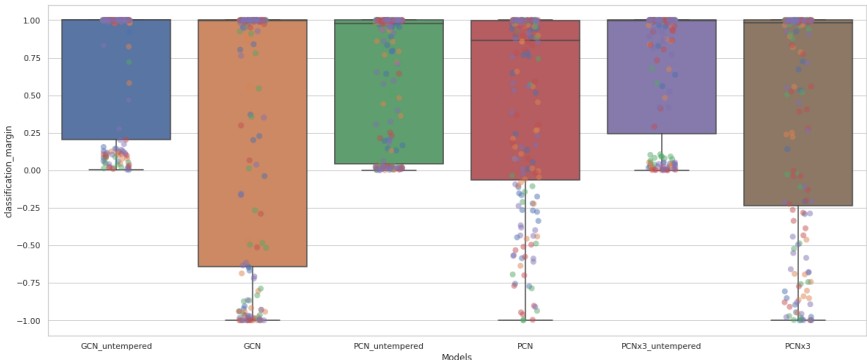

Figure 28: Classification Margin Diagram on influence attack with perturbation equal to the degree of target node and 5 influencing neighboring nodes

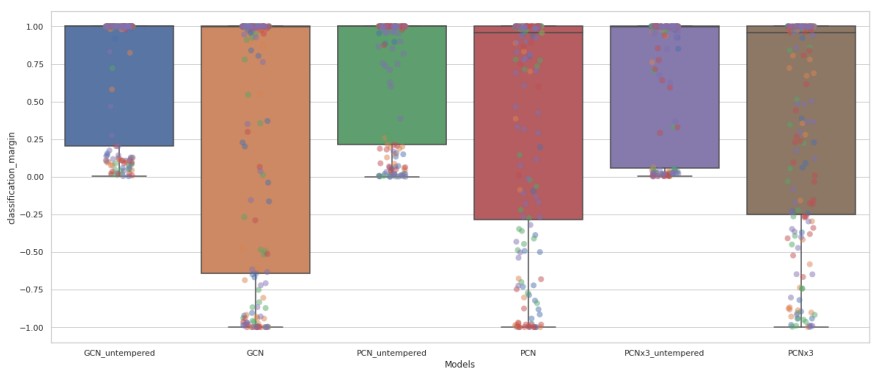

Figure 29: Classification Margin Diagram on on influence attack with perturbation equal to the degree of target node and 5 influencing neighboring nodes

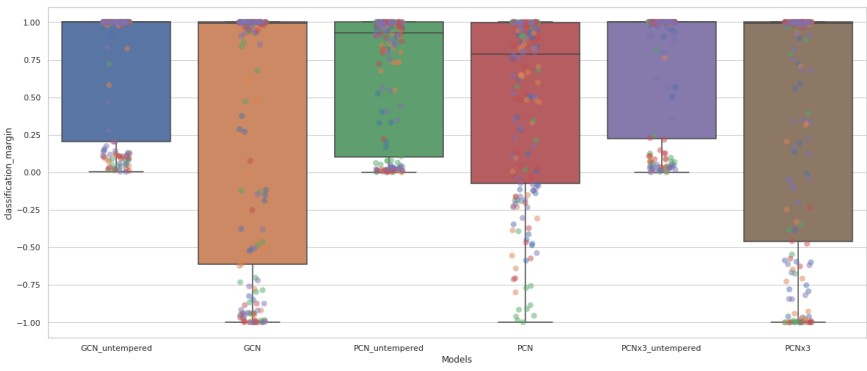

Figure 30: Classification Margin Diagram on influence attack with perturbation equal to 1 and 5 influencing neighboring nodes

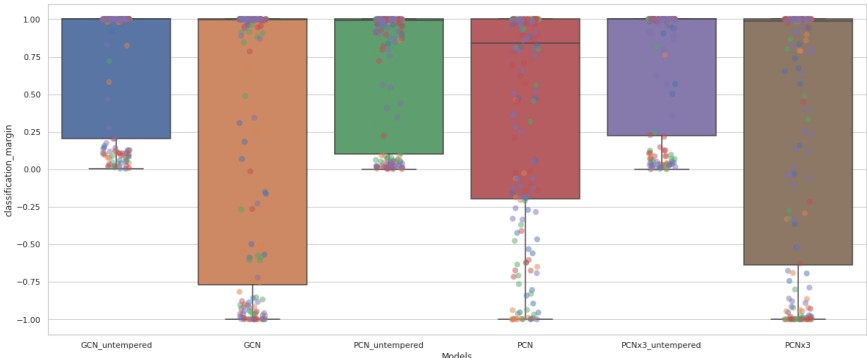

Figure 31: Classification Margin Diagram on on influence attack with perturbation 10 and 5 influencing neighboring nodes

