# OpenReview forum: "Robust Graph Representation Learning via Predictive Coding"
_ICLR.cc/2023/Conference — Submitted to ICLR 2023_

### Official Review · Reviewer_evdf · 2022-10-21

**Confidence:** 3
**Correctness:** 4
**Technical Novelty And Significance:** 2
**Empirical Novelty And Significance:** 2
**Recommendation:** 5

**Clarity, Quality, Novelty And Reproducibility:**

The paper demonstrates limited technical novelty, as it largely reapplies existing work on predictive coding on graphs, onto GCNs. The authors do attempt to show that this may improve robustness against adversarial attacks.

The paper suffers from several issues in clarity and in writing. Below is an incomplete list:
- BP is never defined as “backpropagation”
- In Section 2, the update function is not defined
- In Eq. 4, $x$ should be $h$, and in Eq. 5, $\theta$ should be $w$
- “We follow the formulation of PG graphs” should be “PC”, I believe
- I think the citation next to “GCN” in Table 3 is a mistake
- The model architecture is not entirely clear from the supplement (more in-depth explanation would be nice), although the table of hyperparameters is much appreciated
- There is a broken figure link in Appendix D
- There are many grammatical errors in the writing, which should be fixed

**Strength And Weaknesses:**

This paper applies PC onto GCNs. It is an interesting idea and is a natural next step (after previous work), and there are analyses which quantify robustness. The authors have done comparisons across several tasks and datasets.

Below are some weaknesses which I feel detract from the paper’s main points:
### There is little intuition or exploration on why PC might improve robustness
Although existing works on PC and on classifier energy are cited, it is not well explained why applying PC to models like GCNs might improve robustness against adversarial attacks. The authors offer some minor amount of justification based on the concept of “low-pass filtering” in the abstract, but this is not really expanded upon significantly. It is unclear why one might expect PC to be a reasonable technique for improving robustness (relative to backpropagation).
### The analyses on predictive performance and ECE/MCE are only against severely weakened GCNs
The analyses on predictive performance (Tables 1 and 2) and the reliability diagrams only compare GPCNs (GCNs trained with PC) against vanilla GCNs which are not trained with things like normalization. It is rather well-known that normalization is very important for GCNs, and so these comparisons are rather unfair—the GCNs are severely weakened and so these results are not particularly compelling. It would be more reasonable to compare with a standard GCN trained with dropout/normalization.
### Comparisons in robustness are essentially only against a single method, where other more recent methods exist
Although many methods for improving robustness in GNNs exist (the authors cite several in Section 5), the comparisons on robustness really only compare GPCNs with GCNs, RGCNs, or GATs, and not consistently. In these comparisons, RGCN is the only model I would expect would be a fair baseline for robustness. GCNs and GATs are less meant for robustness, although Table 4 shows that GATs oftentimes outperform GPCNs in terms of robustness anyways.

**Summary Of The Paper:**

The paper applies predictive coding (PC)—an alternative to backpropagation—to graph neural networks, and attempts to show an improvement in robustness to adversarial attacks. The authors describe the predictive-coding scheme for graph convolutional networks (GCNs), and evaluate robustness against several forms of adversarial attacks compared to other GNNs.

**Summary Of The Review:**

The paper offers an interesting application of predictive coding (PC) for GNNs, and attempts to show that GCNs trained with predictive coding rather than traditional backpropagation are more robust against adversarial attacks. Unfortunately, the comparisons in robustness are only done against a single method (i.e. RGCN), even though several other methods exist. Since the models trained with PC are not confidently outperforming existing methods in terms of predictive performance or robustness, this detracts a lot from the main point of the paper. If the paper were to be rewritten more as an exploration of PC on GCNs rather than a novel way to improve robustness, it would be more compelling as a piece of scientific work. Unfortunately, the paper also suffers from numerous issues in terms of clarity and writing, which make it difficult to support its passage.

---

> ### Author Response · Authors · 2022-11-13
> **Answer to reviewer**
>
> We thank the reviewer for the comments.
>
>
> > Why does PC improve robustness?
>
> Robustness is a big problem in deep learning, due to some properties of models trained with backpropagation. For example, these models learn sparse representations. Models trained with PC do not suffer this drawback, as the relaxation process of their energy-based formulation allows the information to spread in the whole network. This leads to models that are better calibrated.
>
> > Analysis provided only against weak baselines
>
> In this work, we have shown that predictive coding models, mostly tested on simple benchmarks such as MNIST, are actually well able to perform tasks on complex benchmarks that are made of graph data. This is not a trivial improvement in the literature of neuroscience-inspired learning methods. In addition, we provide initial evidence that one of such properties is robustness, which is also a novel contribution. This improvement is, however, not yet comparable to the results obtained by standard GNNs, that benefit from ‘tricks’ developed after years of research by the community, such as dropout and batch normalization. It is hence unfair to compare models with a lot of ‘tricks’, or developed specifically to be robust against adversarial attacks, against a simple baseline like ours. It is fair, however, to compare the baseline that we propose against equivalent baselines trained with backpropagation.
>
> > Typos and comments
> We will address them, as well as upload an improved version of the manuscript.

---

### Official Review · Reviewer_3kAN · 2022-10-24

**Confidence:** 3
**Clarity, Quality, Novelty And Reproducibility:** <mentioned in comments above.>
**Correctness:** 2
**Technical Novelty And Significance:** 2
**Empirical Novelty And Significance:** 2
**Recommendation:** 3

**Strength And Weaknesses:**

Pros:
1. This work attempts to expand Predictive coding techniques for multilayer graph neural networks. There has been limited success for PC graphs and this work identifies and extends the predictive coding approach.
2. The choice of experiments and related methods are well chosen.

Comments:
1. Line 3 of Preliminaries section, should it be $X\in \mathbb{R}^{|V|\times d}$ ?
2. In eq.1, what is (t) ? Please define it properly.
3. “Here, the <aggregation> function is a weighted combination of neighbour characteristics with predetermined fixed weights, and aggregate function is a linear transformation.” You meant the <update> function?
4. In eq. 4, what is $x_{u, t}$ ? Shouldn’t it be $h_{u, t}$ ? It seems to me that the notation got mixed up with the referred PC graph paper.
5. (page 3.) typo `PG graphs’? Should be PC graphs.

[PC graph paper]:  “Learning on Arbitrary Graph Topologies via Predictive Coding”

There are lots of typos and errors in the write-up. I have pointed out a few of them. The paper is slightly difficult to follow. I think the contribution of this paper is very incremental in terms of architecture novelty. It is a direct extension of the PC graph paper. The results on adversarial attacks presented are however good and seems promising. I would request the authors to revisit their draft to polish their presentation as IMO the current draft is not ready for the conference.

**Summary Of The Paper:**

This paper address the problem of GNNs being vulnerable to imperceptible adversarial attacks and also has some issues in generalizing out-of-distribution data. The proposed solution, graph predictive coding network (GPCN), uses a novel message-passing scheme developed based on the theory of predictive coding. The authors argue that their approach enhances the robustness of the learned representations. Their experiments show that the representations learned are more robust to adversarial attacks as well as deliver improved results over the GCN counterpart.

**Summary Of The Review:**

There are lots of typos and errors in the write-up. I have pointed out a few of them. The paper is slightly difficult to follow. I think the contribution of this paper is very incremental in terms of architecture novelty. The results on adversarial attacks presented are however good and seems promising. I would request the authors to revisit their draft to polish their presentation as IMO the current draft is not ready for the conference.

---

> ### Author Response · Authors · 2022-11-13
> **Answer to reviewer**
>
> We thank the reviewer for the comments.
> > Typos and errors
>
>
> Thanks; we will upload an improved and polished version of the manuscript that also addresses your comments.
> > Incremental contribution
> The paper that you cite uses predictive coding networks for image generation and classification on the MNIST (and fashionMNIST) dataset. Showing promising results on complex tasks and complex benchmarks that are made of graph data should not be considered an incremental contribution in the literature of neuroscience-inspired learning methods, as it could open up an interesting field of research that studies interesting properties of these types of models on graphs. In this work, we provide initial evidence that one of such properties is robustness, which is also a novel contribution.

---

### Official Review · Reviewer_b3aa · 2022-10-25

**Confidence:** 4
**Correctness:** 2
**Technical Novelty And Significance:** 2
**Empirical Novelty And Significance:** 2
**Recommendation:** 3

**Clarity, Quality, Novelty And Reproducibility:**

I think the use of predictive coding on graphs is novel, it has been tried more often with image processing using convolutional layers. The results overall seem reproducible. However, the attempt at predictive coding does not seem very successful here.

The predictive coding model proposed here could use a schematic.

Table 4: Is GPCN-GCN supposed to be the present study’s architecture? Why are references to the model inconsistent?
What are the +/-
Argument that GPCN outperforms GCN across the board but the confidence intervals (or standard deviation, or whatever is being reported, which should be made clear) are overlapping in over 50% of comparisons.

Figure 3: what do the points represent? What do the different colors of the points represent? Also they are illegible  plotted over the boxplot as presented.

There are typos and awkward phrasings throughout.


**Strength And Weaknesses:**

Overall, I have seen the use of predictive coding as an alternative to backpropagation since PC is more biologically plausible and since it’s easier to parallelize at scale.  The motivation for using PC here doesn’t really come through as they don’t seem to value the biological plausibility or provide benchmarking for efficiency in training.

It does appear to be more robust than the chosen comparators (especially GCNs) on some of the tasks. However, the other formulations (robust GCN and graph attention network) seem to outperform it, which makes me wonder what the real benefit to this model is.

The authors state that GAT and RGCN, which consistently outperform the present method, do so because they are trained with batch normalization and dropout and both use attention mechanisms. Why not try those mechanisms with GPCN? Is there some reason those cannot be used, and if so, isn’t that a limitation of the present model?

**Summary Of The Paper:**

The authors present a predictive coding approach to graph representation learning. The authors show that this approach yields performance that is often worse than GCN and other variants like GAT, but that robustness to structural attacks are better.

**Summary Of The Review:**

This seems to be an interesting idea but not yet mature enough for publication given its performance on chosen tasks.

---

> ### Author Response · Authors · 2022-11-13
> **Answer to reviewer**
>
> We thank the reviewer for the comments.
>
>
> > The motivation for using PC here doesn’t really come through as they don’t seem to value the biological plausibility or provide benchmarking for efficiency in training.
>
> The advantages of PC that you mention (efficiency, parallelization, and biological plausibility) are known advantages of PC that still hold in this case. In this paper, we provide evidence of a new advantage of PC, i.e., that it naturally learns robust representation on graphs. Most of our effort was spent in providing the experiments that show this advantage.
>
> > However, the other formulations (robust GCN and graph attention network) seem to outperform it, which makes me wonder what the real benefit to this model is.
>
>
> The real benefit is that a simple baseline (so, PC applied to GCNs) is able to significantly outperform its counterpart, GCNs trained with BP, in terms of robustness. Furthermore, it obtains competitive performance with specific models that are either developed exactly for robustness (such as robust GCNs), or make use of the attention mechanism. We believe that the fact that specific and advanced models are slightly more robust than our baseline is a nice result, and should not be considered a drawback.
>
> > batch normalization and dropout
>
> They can be used, and lead to an improvement in performance. This improvement is, however, not comparable to the improvement they have on standard GNNs. This is because both dropout and batch normalisation are ‘tricks’ developed after years of research by the machine learning community to improve the results of BP-based models. It is hence unfair to compare models with a lot of ‘tricks’, against a simple baseline like ours.
>
> > Is GPCN-GCN supposed to be the present study’s architecture?
>
> Yes.
>
> > Figure 3: what do the points represent? What do the different colors of the points represent? Also they are illegible plotted over the boxplot as presented.
>
> The points represent the victim nodes selected, while the colours correspond to experiment random seed (each same experiment was repeated 5x)
>
> > Typos, comments on figures, and awkward phrasing.
>
> We will address them, as well as improving the descriptions of the figures, and upload an improved version of the manuscript. If you could point out to the awkward phrases you mention, we will be happy to update them all.

---

### Official Review · Reviewer_opfV · 2022-10-26

**Confidence:** 4
**Correctness:** 3
**Technical Novelty And Significance:** 1
**Empirical Novelty And Significance:** 2
**Recommendation:** 3

**Clarity, Quality, Novelty And Reproducibility:**

The motivation for this study is clear, i.e., to use techniques (such as energy-based design and predictive coding) that have been demonstrated to be effective in enhancing the robustness of deep learning in other more general domains. Graph Predictive Coding is a straightforward method that is easy to follow. However, its novelty is limited because the proposed techniques have been developed in prior work that has not been mined for graph-specific insights.

**Strength And Weaknesses:**

Strength:

1. This paper has a clear motivation and advancing idea. In the introduction section, the authors point out that the robustness of graph neural networks remains a lot of issues to address. Inspired by prior works in other domains, the work presents a new training mechanism on graph neural networks. It finds that the energy-based model and the predictive coding are worth replacing the commonly used backpropagation, then introduces both ideas into message-passing neural networks.

2. The proposed methods establish a non-back propagation approach to train GNNs, which is a technically novel style and different from prior GNNs training works. In detail, Graph Predictive Coding Networks include an energy-based model and predictive coding, which not only minimize the same energy function with two different message-passing mechanisms but also filter the message being passed down from direct neighbors of a particular node.

3. Experimental section is plentiful. Unlike previous work that evaluates the robustness of graph neural networks, this paper measures the test results of the proposed model from a more fundamental perspective (e.g., using confidence to compare robustness). Table 1 also shows that the proposed GCPN achieves the best robustness against perturbations compared with GCN and RGCN.

Weakness:

1. The technical contributions in methods are heavily similar to prior works, making the novelty limited. An idea of predictive coding has been already presented in [1, 2]. Although using the energy-based model and the predictive coding to replace backpropagation is novel to graph learning tasks, the detailed techniques seem like simply adoption on message-passing networks from general machine learning. In addition, the technical description cannot provide new insights for graph neural networks.

[1] An approximation of the error backpropagation algorithm in a predictive coding network with local Hebbian synaptic plasticity, Neural computation.

[2] Learning on arbitrary graph topologies via predictive coding, arXiv.

2. In the experimental comparisons, there are insufficient baselines for comparison. Table 1 compares the proposed GCPN only with the GCN and RGCN. There are, however, numerous other defensive approaches to graph neural networks that may be equally or more effective than RGCN. Since readers may be curious about the efficacy of GCPN in the context of the entire domain, additional baselines need to be added.

3. The number of attack methods used for evaluation is insufficient. Specifically, in Table 4 Poisoning Attack and Section 4.3 Evasion Attack, the authors create adversarial graph samples using only Mettack and Nettack, respectively. Mettack and Nettack are not the only methods for attacking sample generation; many others may be equally or more effective. If the authors provide additional attack methods during evaluation, the defense results of GCPN will demonstrate its superiority more convincingly.


**Summary Of The Paper:**

This paper brings fresh ideas to adversarial graph learning. The proposed methods provide new schemes to train GNN models by predictive coding instead of backpropagation. In particular, Graph Predictive Coding Network modifies predictive coding from general machine learning to message-passing neural networks (GNNs). The design of the experimental section provides a new perspective (using confidence to compare robustness) to show GCPN's better effectiveness on attacked graph samples than other baselines.

**Summary Of The Review:**

The paper is easy-to-follow and the methodology is plainly stated. The proposed techniques are an early attempt at graph tasks. To evaluate GPCN's robustness, the experiments are pretty comprehensive from various dimensions: poisoning and evasion, global and targeted, direct and indirect. However, because both the energy-based model and predictive coding that makeup GPCN are existing technologies, the originality, and novelty of the paper are limited. Overall, I tend to reject this submission.

---

> ### Author Response · Authors · 2022-11-13
> **Answer to Reviewer**
>
> We thank the reviewer for the comments.
>
> > Although using the energy-based model and the predictive coding to replace backpropagation is novel to graph learning tasks, the detailed techniques seem like simply adoption on message-passing networks from general machine learning.
>
> The papers that you have cited study the properties of predictive coding networks for image generation and classification on the MNIST (and fashionMNIST) dataset. Showing promising results on complex tasks and complex benchmarks that are made of graph data is not at all a trivial improvement in the literature of neuroscience-inspired learning methods. Hence, we believe that providing evidence that predictive coding networks are able to obtain competitive performance on graph neural networks is an important contribution, as it could open up an interesting field of research that studies interesting properties of these kinds of models on graphs. In this work, we provide initial evidence that one such property is robustness, which is also a novel contribution.
>
>
> > In the experimental comparisons, there are insufficient baselines for comparison. Table 1 compares the proposed GCPN only with the GCN and RGCN.
>
> Note that the goal of this work is to provide initial evidence that the theory of predictive coding can be used to train graph neural networks that are robust and well calibrated. To this end, we test a simple baseline of GPCNs against simple baselines trained with backpropagation. We do not expect our method to be comparable against state-of-the-art models, developed with the specific goal of being robust, and it is not the goal of the paper. Our work, however, shows that GPCNs have a natural tendency of learning robust representations, and that they could hence be considered when developing models suited for critical tasks.
>
> > The number of attack methods used for evaluation is insufficient.
>
> We will provide additional evidence in the following days.

---

### Decision · Program_Chairs · 2023-01-20

**Decision:**

Reject

**Justification For Why Not Higher Score:**

As mentioned, there are several weaknesses that need to be addressed in a revised version.

**Justification For Why Not Lower Score:**

N/A

**Metareview: Summary, Strengths And Weaknesses:**

There was a clear consensus that the paper addresses an interesting problem setting. However, the reviewers also agreed on various negative points including incremental technical contribution, lack of presentation, and insufficient comparisons. The author feedback did not clarify these points sufficiently. The authors are encouraged to revise the paper.